# VISION-LSTM: xLSTM AS GENERIC VISION BACKBONE

**Benedikt Alkin**[1,2] **Maximilian Beck**[1,3] **Korbinian Pöppel**[1,3]
**Sepp Hochreiter**[1,2,3] **Johannes Brandstetter**[1,2]
[1]ELLIS Unit Linz, Institute for Machine Learning, JKU Linz, Austria
[2]Emmi AI GmbH, Linz, Austria [3]NXAI GmbH, Linz, Austria
{alkin,brandstetter}@ml.jku.at

## ABSTRACT

Transformers are widely used as generic backbones in computer vision, despite initially introduced for natural language processing. Recently, the Long Short-Term Memory (LSTM) has been extended to a scalable and performant architecture – the xLSTM – which overcomes long-standing LSTM limitations via exponential gating and parallelizable matrix memory structure. In this paper, we introduce Vision-LSTM (ViL), an adaption of the xLSTM building blocks to computer vision. ViL comprises a stack of xLSTM blocks where odd blocks process the sequence of patch tokens from top to bottom while even blocks go from bottom to top. ViL achieves strong performances on classification, transfer learning and segmentation tasks as well as a beneficial pre-training cost-to-performance trade-off. Experiments show that ViL holds promise to be further deployed as new generic backbone for computer vision architectures.
Project page: https://nx-ai.github.io/vision-lstm/

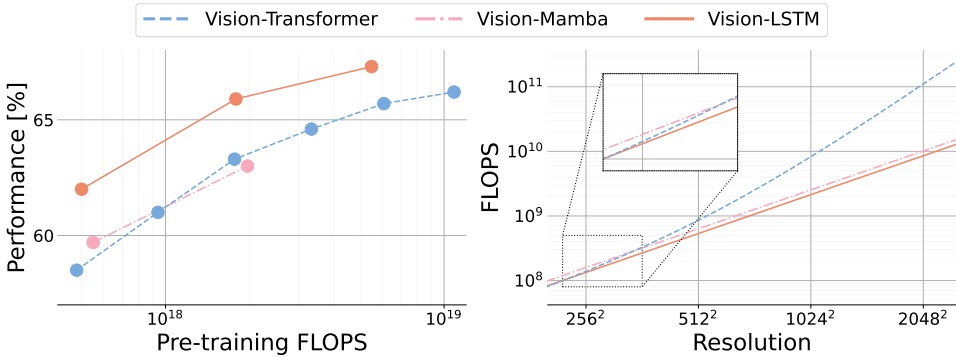

Figure 1: The efficient and scalable design of Vision-LSTM shows strong performances, uses less FLOPS than Transformer/Mamba counterparts and scales linear to higher resolutions. Performance is averaged over ImageNet accuracy, ADE20K mIoU and VTAB-1K accuracy.

## 1 INTRODUCTION

Language modeling architectures — such as Transformers (Vaswani et al., 2017; Achiam et al., 2023; Team et al., 2023) or more recently State Space Models (Gu et al., 2021; Gupta et al., 2022) such as Mamba (Gu & Dao, 2023) — are commonly adapted to the domain of computer vision to make use of their powerful modeling capabilities. However, in natural language processing, an input sentence is typically encoded into tokens that represent words or common subwords (Bostrom & Durrett, 2020) via a discrete vocabulary. To encode images into a set of tokens, Vision Transformer (Dosovitskiy et al., 2021) (ViT) proposed to group an input image into non-overlapping patches (of e.g. 16x16 pixel), linearly project them into a sequence of so-called patch tokens and add positional information to these tokens. This sequence can then be processed by language modeling architectures.

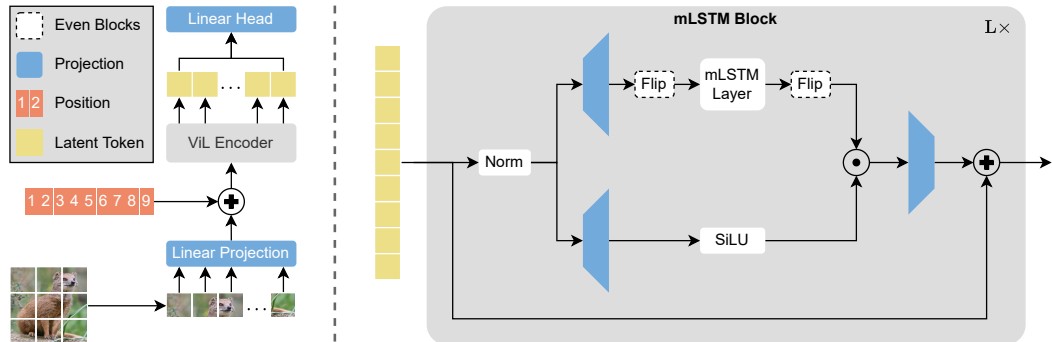

Figure 2: Schematic overview of Vision-LSTM (ViL). Following ViT (Dosovitskiy et al., 2021), an input image is split into patches and linearly projected. Then, a learnable vector is added per position to the patches, producing a sequence of patch tokens. This sequence is then processed by alternating mLSTM blocks where even blocks flip the sequence before and after the mLSTM layer. For classification, ViL uses the concatenation of the first and the last patch as input to a linear classification head. ViL is an isotropic architecture, i.e., all blocks have the same input and output dimension and no downsampling layers are used except the initial patch embedding. Projection layers process each patch individually and the mLSTM exchanges information between patches.

The Extended Long Short-Term Memory (xLSTM) family (Beck et al., 2024) was recently introduced as a new architecture for language modeling. It demonstrates the resurgence of LSTM in the LLM era, performing favorably against the likes of Transformers and State Space Models (SSMs). Analogous to existing vision versions of Transformers or SSMs, e.g., ViT (Dosovitskiy et al., 2021) or Vision Mamba (Zhu et al., 2024), which have produced great results in various computer vision tasks (Singh et al., 2023; Kirillov et al., 2023; Oquab et al., 2023; Peebles & Xie, 2023; Alkin et al., 2024b), we introduce Vision LSTM (ViL) – a generic computer vision backbone that uses xLSTM blocks as its core components. To adjust xLSTM (an autoregressive model) to computer vision (an often non-autoregressive domain), we employ a stack of alternating mLSTM blocks (Beck et al., 2024) where odd blocks process patches row-wise from top left to bottom right and even blocks go from bottom right to top left. This simple alternating design allows ViL to efficiently process non-sequential inputs, such as images, without introducing additional computations.

Similar to vision adaptions of SSMs (Liu et al., 2024; Zhu et al., 2024; Wang et al., 2024), ViL can exhibit linear computational and memory complexity w.r.t. sequence length which makes it appealing for tasks that benefit from high-resolution images such as medical imaging (Chen et al., 2021; Hatamizadeh et al., 2022; Valanarasu et al., 2021; Xu et al., 2024), segmentation (Kirillov et al., 2023; Cheng et al., 2022), or physics simulations (Bi et al., 2023; Nguyen et al., 2023; Bodnar et al., 2024; Alkin et al., 2024a). In contrast, ViT's computational complexity scales quadratically due to the self-attention mechanism, rendering them costly to apply to high-resolution tasks.

Our contributions summarize as follows:

- We introduce Vision-LSTM (ViL), an adaption of the mLSTM to computer vision tasks that can serve as a generic vision backbone with linear complexity.

- We show modeling capacity and generalization in the common vision benchmark of pre-training models on ImageNet-1K, followed by fine-tuning on transfer classification and semantic segmentation tasks.

- We ablate various architectural design choices to evaluate their impact on performance and provide insights into the model design.

- We discuss potential future directions and current limitations that, once addressed, will improve ViL even further.

## 2 METHOD

Vision-LSTM (ViL) introduces xLSTM (Beck et al., 2024) to computer vision, similar to other vision adaptions of sequence modeling architectures, e.g., Vision Transformers (Dosovitskiy et al., 2021), Vision Mamba (Zhu et al., 2024), or Vision RWKV (Duan et al., 2024).

### 2.1 PRELIMINARIES

In the notation of sequence modeling, we consider a series of input vectors $\boldsymbol{x}_t \in \mathbb{R}^D$. This series is created by reshaping an image $\tilde{\mathbf{X}} \in \mathbb{R}^{H_I \times W_I \times C_{\text{in}}}$ into a sequence of flattened 2D patches $\bar{X} \in \mathbb{R}^{T \times (H_P \cdot W_P \cdot C_{\text{in}})}$ and then projected to $\boldsymbol{X} \in \mathbb{R}^{T \times D}$ via a shared linear projection. $D$ is the hidden dimension, $(H_I, W_I)$ is the image resolution, $C_{\text{in}}$ is the number of image channels, $T$ is the number of patches and $(H_P, W_P)$ is the patch size. After creating a sequence of patches, ViL iteratively refines the features of the patch sequence by processing it with a stack of mLSTM blocks where the sequence is flipped within every second block.

The key innovations of the mLSTM (Beck et al., 2024) are the enhanced storage capacity compared to the classical LSTM (Hochreiter & Schmidhuber, 1997) by using a matrix memory cell $\boldsymbol{C} \in \mathbb{R}^{d \times d}$ instead of a scalar memory cell $c \in \mathbb{R}$ and introducing exponential gates (instead of sigmoid gates) to the input and forget gates, where $d$ is the hidden dimension within the mLSTM block (typically $d = 2D$).

Intuitively, the mLSTM is a more expressive and faster version of the classical LSTM that can be efficiently parallelized on modern hardware. In ViL, the mLSTM is used to process dependencies between patches, similar to how the attention exchanges information between patches in a ViT. The mLSTM is embedded into a gated MLP architecture, as shown on the right of Figure 2, where the weight matrices of the MLP process each patch individually and the mLSTM exchanges information between patches. For completeness, we outline the forward pass of the mLSTM in the following paragraphs.

The mLSTM (Beck et al., 2024) is a recurrent neural network, which maps a state $(\boldsymbol{h}_{t-1}, \boldsymbol{C}_{t-1}, \boldsymbol{n}_{t-1})$ to a successor state $(\boldsymbol{h}_t, \boldsymbol{C}_t, \boldsymbol{n}_t)$ given input $\boldsymbol{x}_{t-1}$. Thereby, $\boldsymbol{h}_t \in \mathbb{R}^d$ denotes the hidden state, $\boldsymbol{C}_t \in \mathbb{R}^{d \times d}$ is the cell state and $\boldsymbol{n}_t \in \mathbb{R}^d$ corresponds to a normalizer state. The full forward pass of the mLSTM is as follows (Beck et al., 2024):

$$
\begin{aligned}
\boldsymbol{C}_t &= f_t\, \boldsymbol{C}_{t-1} \,+\, i_t\, \boldsymbol{v}_t\, \boldsymbol{k}_t^\top && \text{cell state} & (1) \\
\boldsymbol{n}_t &= f_t\, \boldsymbol{n}_{t-1} \,+\, i_t\, \boldsymbol{k}_t && \text{normalizer state} & (2) \\
\boldsymbol{h}_t &= \boldsymbol{o}_t \odot \tilde{\boldsymbol{h}}_t \qquad \tilde{\boldsymbol{h}}_t = \boldsymbol{C}_t \boldsymbol{q}_t / \max\left\{|\boldsymbol{n}_t^\top \boldsymbol{q}_t|, 1\right\} && \text{hidden state} & (3) \\
\boldsymbol{q}_t &= \boldsymbol{W}_q\, \boldsymbol{x}_t \,+\, \boldsymbol{b}_q && \text{query input} & (4) \\
\boldsymbol{k}_t &= \frac{1}{\sqrt{d}} \boldsymbol{W}_k\, \boldsymbol{x}_t \,+\, \boldsymbol{b}_k && \text{key input} & (5) \\
\boldsymbol{v}_t &= \boldsymbol{W}_v\, \boldsymbol{x}_t \,+\, \boldsymbol{b}_v && \text{value input} & (6) \\
i_t &= \exp(\tilde{i}_t) \qquad \tilde{i}_t = \boldsymbol{w}_i^\top\, \boldsymbol{x}_t \,+\, b_i && \text{input gate} & (7) \\
f_t &= \exp(\tilde{f}_t) \qquad \tilde{f}_t = \boldsymbol{w}_f^\top\, \boldsymbol{x}_t \,+\, b_f && \text{forget gate} & (8) \\
\boldsymbol{o}_t &= \sigma(\tilde{f}o_t) \qquad \tilde{\boldsymbol{o}}_t = \boldsymbol{W_o}\, \boldsymbol{x}_t \,+\, \boldsymbol{b_o} && \text{output gate} & (9)
\end{aligned}
$$

As exponential activation functions can lead to large activations, the input and forget gates are stabilized with an additional state $m_t$:

$$
\begin{aligned}
m_t &= \max\left(\log(f_t) + \boldsymbol{m}_{t-1}, \log(f_t)\right) && \text{stabilizer state} & (10) \\
i'_t &= \exp\left(\log(i_t) - m_t\right) = \exp\left(\tilde{i} - m_t\right) && \text{stabilized input gate} & (11) \\
f'_t &= \exp\left(\log(f_t) + m_{t-1} - m_t\right) && \text{stabilized forget gate} & (12)
\end{aligned}
$$

As the mLSTM has no memory mixing, i.e, interactions between hidden states from one timestep to the next, it can be fully parallelized for fast computation on modern hardware. For a detailed discussion and theory of the cell state update, further details to the mLSTM we refer to the original work (Beck et al., 2024).

## 2.2 VISION-LSTM (VIL)

Vision-LSTM (ViL) is a generic backbone for computer vision tasks, which is residually built from mLSTM blocks, as visualized in Figure 2. Following ViT (Dosovitskiy et al., 2021), ViL first splits an image into non-overlapping patches via a shared linear projection, then adds learnable positional embeddings to each patch token. At the core of ViL are alternating mLSTM blocks, which are fully parallelizable and equipped with a matrix memory combined with a covariance update rule. Odd mLSTM blocks process patch tokens from top left to bottom right while even blocks go from bottom right to top left.

Formally, the forward pass of a pair of ViL blocks is:

$$Y' = X + \text{Block}_\theta(X) \tag{13}$$
$$Y = Y' + \text{Flip}(\text{Block}_\phi(\text{Flip}(Y'))) \tag{14}$$

Where "Flip" reverses the sequence and "Block$_\theta$" and "Block$_\phi$" corresponds to mLSTM blocks with parameters $\theta$ and $\phi$ (shown in Figure 2, right).

A key motivation of ViL is that the autoregressive mLSTM can operate in a recurrent, parallel or chunkwise mode, each with distinct FLOPS and runtime characteristics. Given a sequence length $T$ and hidden dimension $d$, the complexity of the recurrent mode is $\mathcal{O}(Td^2)$ and needs to be processed sequentially, whereas the parallel mode has complexity $\mathcal{O}(T^2d)$ and is fully parallelizable. The chunkwise mode combines the advantages of the other modes by introducing a chunksize $S$ where the parallel mode is used within chunks and the recurrent mode between chunks. This allows high parallelization, minimal operations and linear scaling with $T$. Complexity wise, the chunkwise mode has $\mathcal{O}(\frac{T}{S}S^2d + \frac{T}{S}d^2)$ or $\mathcal{O}(TSd + \frac{T}{S}d^2)$ where $\frac{T}{S}$ corresponds to the number of chunks.

## 3 EXPERIMENTS

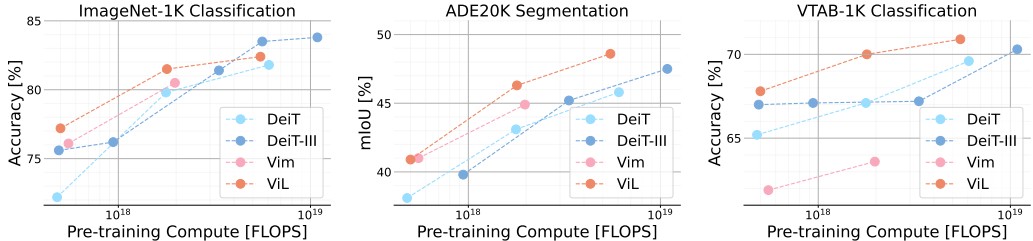

Figure 3: Performance overview of ImageNet-1K pre-trained models in relation to pre-training compute. ViL shows strong performances across classification (ImageNet-1K), semantic segmentation (ADE20K) and transfer classification (VTAB-1K) tasks.

We pre-train models on ImageNet-1K (Deng et al., 2009), which contains 1.3M training images and 50K validation images where each image belongs to one of 1000 classes. ViL models are trained for 800 epochs (tiny) or 400 epochs (small, base) on 192x192 resolution with a learning rate of 1e-3 using a cosine decay schedule. Afterwards, the model is fine-tuned on 224x224 resolution for 20 epochs using a learning rate of 1e-5. Detailed hyperparameters can be found in Appendix Table 10.

We then transfer the pre-trained models to serveral benchmark tasks: ImageNet-1K classification on the validation set, ADE20K (Zhou et al., 2019) semantic segmentation and VTAB-1K (Zhai et al., 2019) classification. These benchmarks evaluate global image understanding (ImageNet-1K), semantic local and global understanding (ADE20K) and few-shot generalization to a diverse set of

Table 1: ImageNet-1K pre-training accuracy. All models use a patch size of 16x16 with 224x224 resolution at most. Models with "+" in their "Epochs" column pre-train on lower resolution followed by fine-tuning on 224x224 resolution for some epochs. ViL performs favorably against an isotropic convolutional architecture (ConvNeXt) and vision adaptions of transformers (DeiT series), RWKV (VRWKV) and Mamba (Vim, Mamba®). Appendix Table 9 confirms these results on OOD and robustness evaluations of these classifiers.

| Model | Epochs | #Params | FLOPS | IN-1K |
|---|---|---|---|---|
| DeiT-T (Touvron et al., 2021a) | 300 | 6M | 1.3G | 72.2 |
| DeiT-II-T (Touvron et al., 2022a) | 400 | 6M | 1.3G | 73.5 |
| DeiT-III-T (reimpl.) | 800+20 | 6M | 1.3G | 76.2 |
| VRWKV-T (Duan et al., 2024) | 300 | 6M | 1.2G | 75.1 |
| Vim-T (Zhu et al., 2024) | 300 | 7M | 1.5G | 76.1 |
| Mamba®-T (Wang et al., 2024) | 280+20 | 9M | 1.6G | 77.4 |
| ViL-T | 800+20 | 6M | 1.3G | **78.3** |
| DeiT-S (Touvron et al., 2021a) | 300 | 22M | 4.6G | 79.8 |
| DeiT-II-S (Touvron et al., 2022a) | 400 | 22M | 4.6G | 80.7 |
| DeiT-III-S (Touvron et al., 2022b) | 800+20 | 22M | 4.6G | 81.4 |
| ConvNeXt-S (*iso.*) (Liu et al., 2022) | 300 | 22M | 4.3G | 79.7 |
| VRWKV-S (Duan et al., 2024) | 300 | 24M | 4.6G | 80.1 |
| Vim-S (Zhu et al., 2024) | 300 | 26M | 5.3G | 80.5 |
| Mamba®-S (Wang et al., 2024) | 280+20 | 28M | 5.5G | 81.1 |
| ViL-S | 400+20 | 23M | 4.7G | **81.5** |
| DeiT-B (Touvron et al., 2021a) | 300 | 86M | 17.6G | 81.8 |
| DeiT-II-B (Touvron et al., 2022a) | 400 | 86M | 17.6G | 82.7 |
| DeiT-III-B (Touvron et al., 2022b) | 800+20 | 86M | 17.6G | **83.7** |
| ConvNeXt-B (*iso.*) (Liu et al., 2022) | 300 | 87M | 16.9G | 82.0 |
| VRWKV-B (Duan et al., 2024) | 300 | 94M | 18.2G | 82.0 |
| Mamba®-B (Wang et al., 2024) | 280+20 | 99M | 20.6G | 82.9 |
| ViL-B | 400+5 | 89M | 17.9G | 82.4 |

19 VTAB-1K classification datasets, which include natural images, specialized imagery (medical and satellite) and structured tasks (camera angle prediction, depth estimation, object counting, . . . ).

Figure 3 shows an overview of performance metrics in relation to total pre-training compute where ViL performs favorably against heavily optimized transformer protocols (DeiT, DeiT-III) and Vision Mamba (Vim). Detailed results are presented in the following sections.

As ViTs are well established in the vision community, they underwent multiple optimization cycles over the years (Dosovitskiy et al., 2021; Touvron et al., 2021a; 2022a; 2021b; 2022b). Therefore, a vast part of the hyperparameter space for pre-training ViTs has been explored. Since this work is the first to apply xLSTM to computer vision, considerably less effort has been put into hyperparameter tuning and architecture optimization, suggesting that future work could improve ViL even further.

### 3.1 IMAGENET-1K CLASSIFICATION

Table 1 relates parameter counts and FLOPS to validation accuracy after pre-training on ImageNet-1K. ViL outperforms heavily optimized ViT protocols and other backbones on the tiny and small scale. While ViL does not outperform all other models on the base scale, evaluations on downstream tasks (as shown later in Table 2 and Table 3) show that ViL-B still learns strong features, particularly for semantic segmentation and structured tasks.

### 3.2 ADE20K SEMANTIC SEGMENTATION

Table 2 shows results for transferring ImageNet-1K pre-trained models to ADE20K (Zhou et al., 2019) semantic segmentation using UperNet (Xiao et al., 2018). Also here, ViL shows strong performances across the board, even outperforming DeiT-III-B despite the lower ImageNet-1K accuracy of ViL-B. The high resolution of the ADE20K segmentation task (512x512) results in a total of 1024 patch tokens where the quadratic complexity of self-attention is significantly more expensive than the linear complexity of the mLSTM, resulting in much fewer FLOPS for ViL. Additionally, the

efficient alternating block design results in lower FLOPS than Mamba-based vision models (which also have linear complexity).

Table 2: Semantic segmentation results on ADE20K (Zhou et al., 2019) using UperNet (Xiao et al., 2018). We report mean intersection over union (mIoU) and pixelwise accuracy (ACC) for single- and multi-scale evaluation. Models are trained for 160K updates with a batchsize of 16 on 512x512 resolution. We use a feature pyramid consisting of rescaled feature maps after the 4th, 6th, 8th and final block. Detailed hyperparameters are listed in Appendix Table 12. FLOPS are calculated only from the backbone at 512x512 resolution as all models use the same segmentation head.

| Model | #Params | FLOPS | Single-scale | | Multi-scale | |
|---|---|---|---|---|---|---|
| | | | mIoU | ACC | mIoU | ACC |
| DeiT-T | 10M | 10.4G | 38.1 | 78.2 | 40.3 | 79.9 |
| DeiT-III-T | 10M | 10.4G | 39.8 | 79.2 | 42.2 | 80.7 |
| Vim-T | 13M | 7.7G | 41.0 | - | - | - |
| ViL-T | 11M | **6.6G** | **41.2** | **80.2** | **43.1** | **81.3** |
| DeiT-S | 41M | 31.7G | 43.1 | 80.7 | 45.2 | 81.8 |
| DeiT-III-S | 41M | 31.7G | 45.2 | 81.5 | 46.3 | 82.3 |
| Vim-S | 46M | 27.3G | 44.9 | - | - | - |
| Mamba®-S | 56M | 27.6G | 45.3 | - | - | - |
| ViL-S | 42M | **24.4G** | **46.3** | **82.0** | **47.9** | **82.9** |
| DeiT-B | 113M | 107.0G | 45.8 | 82.1 | 47.0 | 82.9 |
| DeiT-III-B | 113M | 107.0G | 47.5 | 82.6 | 49.0 | 83.3 |
| Mamba®-B | 132M | 102.8G | 47.7 | - | - | - |
| ViL-B | 115M | **93.6G** | **48.6** | **82.8** | **49.6** | **83.3** |

## 3.3 VTAB-1K Transfer Classification

Table 3: Transfer classification accuracies on the VTAB-1K (Zhai et al., 2019) benchmark using ImageNet-1K pre-trained models. VTAB-1K consists of 19 datasets split into 7 natural, 4 specialized and 8 structured datasets. We show averages per category and the average accuracy over all 19 datasets (Appendix Table 8 lists all individual accuracies). ViL shows strong generalization performance, outperforming heavily optimized ViT protocols and Vim on the full VTAB-1K benchmark. ViL performs exceptionally well on the structured category. We tune the learning rate for each model and dataset on the validation set and report the average testset accuracy over 5 seeds. Appendix Table 11 lists further hyperparameters.

| Model | #Params | FLOPS | Natural | Specialized | Structured | Average |
|---|---|---|---|---|---|---|
| DeiT-T | 6M | 1.3G | 69.2 | 82.0 | 53.3 | 65.2 |
| DeiT-III-T | 6M | 1.3G | 71.9 | 82.6 | 55.2 | 67.1 |
| Vim-T | 7M | 1.5G | 68.0 | 80.7 | 47.1 | 61.9 |
| ViL-T | 6M | 1.3G | **73.6** | **83.4** | **56.1** | **68.3** |
| DeiT-S | 22M | 4.6G | 73.3 | 83.8 | 53.2 | 67.1 |
| DeiT-III-S | 22M | 4.6G | 75.0 | 83.2 | 52.3 | 67.2 |
| Vim-S | 26M | 5.3G | 69.6 | 81.7 | 49.4 | 63.6 |
| ViL-S | 23M | 4.7G | **75.3** | **84.3** | **58.3** | **70.0** |
| DeiT-B | 86M | 17.6G | 76.5 | **85.2** | 55.7 | 69.6 |
| DeiT-III-B | 86M | 17.6G | **77.6** | 84.8 | 56.6 | 70.3 |
| ViL-B | 89M | 17.9G | 76.6 | 84.7 | **59.1** | **70.9** |

Table 3 shows transfer classification results for ImageNet-1K pre-trained models on the VTAB-1K (Zhai et al., 2019) benchmark. VTAB-1K consists of 19 datasets split into 7 natural datasets (such as CIFAR100 (Krizhevsky, 2009) or Caltech101 (Fei-Fei et al., 2006)), 4 specialized datasets (medical imaging (Veeling et al., 2018; Kaggle & EyePacs, 2015) and remote sensing (Helber et al., 2019; Cheng et al., 2017)) and 8 structured datasets (with tasks such as object counting (Johnson et al., 2017) or binned depth estimation (Geiger et al., 2013)). We follow common practices and tune the learning rate per model and dataset on the validation set followed by training with the best learning rate on the union of train and validation set. The performance metric is the average testset accuracy over 5 seeds. ViL shows strong transfer classification performance outperforming all other

models on the average over all 19 datasets. ViL performs particularly well on the structured datasets where ViL-B outperforms DeiT-III-B despite ViL-B having lower ImageNet-1K accuracy.

## 4  ABLATION STUDIES

We ablate various design choices of ViL by training ViL-T models for 100 epochs on ImageNet-1K in 224x224 resolution, other hyperparameters follow the ones from Section 3 (see also Appendix B.3). We then report the validation accuracy on ImageNet-1K and fine-tune the model on ADE20K to ensure that design choices are not overfitted to classification. We also use a reduced segmentation pipeline where we use a linear segmentation head and train for 40K updates using a batch size of 16 (other hyperparameters follow Appendix 12).

### 4.1  ARCHITECTURAL DESIGN

We consider various architecture design choices in Table 4.

Table 4: Architecture design ablation studies.  Default settings

(a) **Traversal Directions**

| Directions | IN1K | ADE20K |
|---|---|---|
| Uni-dir. | 72.2 | 28.6 |
| Bi-dir. | 73.7 | 31.7 |
| Quad-dir. | **73.8** | **33.1** |
| Oct-dir. | 73.5 | 32.4 |

(b) **QK Convolution**

| Convolution | IN1K | ADE20K |
|---|---|---|
| None | 72.3 | 29.2 |
| Causal-Conv1D | 72.8 | 27.8 |
| Conv1D | 72.8 | 28.4 |
| Conv2D | **73.7** | **31.7** |

(c) **Positional Embedding**

| Pos. Embed. | IN1K | ADE20K |
|---|---|---|
| ✗ | **73.7** | 31.0 |
| ✓ | **73.7** | **31.7** |

(d) **Concurrency**

| Concurrency | IN1K | ADE20K |
|---|---|---|
| Sequential | **73.7** | **31.7** |
| Parallel | 73.0 | 30.6 |

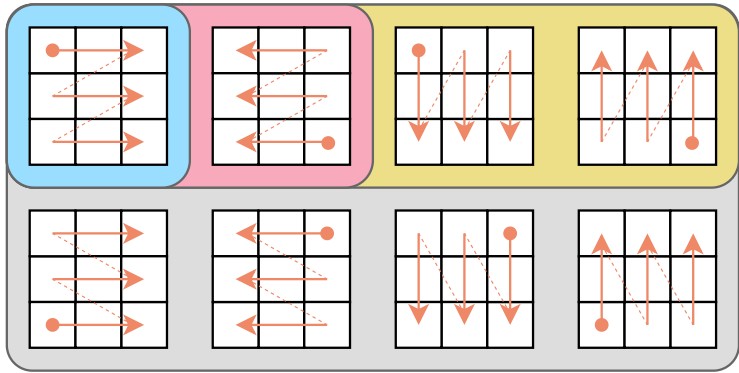

Figure 4:  Uni-directional ,  bi-directional ,  quad-directional  and  oct-directional  traversal paths. Squares represent individual patch tokens. Traversal starts at the circle and goes in direction of the arrow, if no further patches are in a row/column, the traversal continues in the next row/column as indicated by the dashed line.

**(a) Traversal Directions**  Traversing the sequence in at least two directions greatly improves performance due to the non-causal 2D structure of images. Adding column-wise traversal directions (Quad-dir.) could even further improve semantic segmentation performance. Additionally using 4 instead of 2 starting positions (Oct-dir.) shows no benefit. Note that all variants have the same amount of FLOPS due to sequential application of different directions. Directions are visualized in Figure 4.

We use "Bi-dir." for our final models due to current technical limitations which would slow down training on more than 2 directions. This limitation comes from the current lack of optimized hardware implementations of the mLSTM (e.g., CUDA kernels) where we instead rely on `torch.compile`, a generic speed optimization method from PyTorch (Paszke et al., 2019), to optimize computations. Our implementation of quad- and oct-directional traversals is not compatible with `torch.compile`, which results in approximately double the runtime. We therefore train all models from Section 3 with "Bi-dir." only. Note that this is only a technical limitation, not a methodical one and the ablation study suggest that future ViL models could be even better using a quad-directional design.

**(b) QK Convolution** The mLSTM block design uses a causal 1D convolution to aggregate local context to improve storage/retrieval to/from the cell state $C$. This is done by applying a convolution layer to $X$ before projecting it to $Q$ with $W_q$ and $K$ with $W_k$ respectively. The convolution is shared for $Q$ and $K$. The causal 1D structure of the convolution from the original mLSTM (Beck et al., 2024) is necessary due to the causal 1D structure of language modeling. However, as images are neither causal nor 1D structures, we replace the causal 1D convolution with a 2D convolution (with kernel size 3). This allows the mLSTM to make better storage/retrieval decisions through the added local context.

**(c) Positional Embedding** ViTs require positional embedding to tell the model where each patch is located in the image, suffering heavy performance losses if the position is not required (Dosovitskiy et al., 2021; Chu et al., 2023). The mLSTM is an autoregressive model, which makes it optional to add positional embeddings as it can recognize the position of the current patch based on how many patches have been processed. However, the ablation shows that it is nevertheless beneficial to provide this information explicitly as it improves segmentation results without hurting classification performance.

**(d) Sequential vs. Parallel** Related architectures use a parallel design where a sequence is processed from multiple directions in a single block (Zhu et al., 2024; Duan et al., 2024). We investigate a similar design where we apply both directions in parallel instead of sequentially. To keep parameters and FLOPS constant, we apply the directions akin to parallel transformer blocks (Wang, 2021) while halving the depth.

$$Y = X + \text{Block}_\theta(X) + \text{Flip}(\text{Block}_\phi(\text{Flip}(X))) \tag{15}$$

## 4.2 CLASSIFICATION DESIGN

In order to perform classification from a sequence of tokens, it is common to aggregate information from the whole sequence, which is then used as input to a classification head. The most common methods to do this aggregation are (i) adding a learnable [CLS] token to the input sequence or (ii) averaging all patch tokens to produce an [AVG] token. In ViTs, whether to use the [CLS] or [AVG] token is typically a hyperparameter, where both variants achieve comparable performances. On the contrary, other sequence models models often require specialized classification designs. For example, Vim (Zhu et al., 2024) requires the [CLS] token to be in the middle of the sequence, suffering heavy performance losses if other classification designs, e.g., an [AVG] token or two [CLS] tokens at start and end of the sequence, are employed.

We explore different classification designs for ViL in Table 5. (a) We choose concatenating the first and last patch as aggregation method due to its strong classification performance. As our final models also perform well in semantic segmentation (see Table 2), we do not retrain models with [AVG] aggregation even though the ablation suggests that this could boost performance even further for segmentation tasks. (b) Adding learnable [CLS] tokens show no benefit. Therefore, we do not use any [CLS] tokens for ViL.

## 5 LIMITATIONS AND FUTURE WORK

The biggest limitation of ViL is the current lack of an optimized hardware implementation of the mLSTM, which results in longer runtimes than ViTs, which have multiple optimized hardware im-

Table 5: Classification design. (a) ViL aggregates classification information well in the first and the last patches (bilateral), leading to good classification performance if the first and last patches are averaged or concatenated. Averaging all patches ([AVG]) or the 4 center patches (Center [AVG]) results in strong segmentation performances but lackluster classification performances. (b) Adding learnable [CLS] tokens to the start and end of the input sequence (Bilateral [CLS]) offers no benefit over simply using the first and the last patch. Incorporating a [CLS] token in the middle of the sequence, akin to Vim (Zhu et al., 2024), does not improve performance. Default settings

(a) **Patch-based Aggregation**

| Aggregation | IN1K | ADE20K |
|---|---|---|
| Bilateral Mean | 73.0 | 31.5 |
| Bilateral Concat | **73.7** | 31.7 |
| [AVG] | 72.6 | **32.8** |
| Center [AVG] | 72.4 | 32.1 |

(b) **[CLS]-based Aggregation**

| Aggregation | IN1K |
|---|---|
| Concat Bilateral Patches | **73.7** |
| Mid [CLS] | 71.8 |
| Bilateral [CLS] | 73.5 |
| Mid + Bilateral [CLS] | 73.0 |

plementations (Dao et al., 2022; Dao, 2023). This makes a runtime/throughput analysis of models, a vital metric to judge practicability, difficult as the practical relevance of inefficient implementations is quite low. As a proxy, we report FLOP counts, where ViL is comparable to ViT on low-resolution tasks and far better than ViT on high-resolution tasks due to its linear complexity. While FLOPS are far from an optimal proxy for runtime/throughput, they suggest that ViL can be much faster than ViT on high-resolution tasks once an optimized hardware implementation exists. Note that ViL is already faster than Vim (see Appendix A.1) despite its optimized hardware implementation.

This limitation snowballs in multiple other directions. For example, scaling model size further, tuning hyperparameters, training on larger datasets, exploring self-supervised pre-training or investigating hierarchical architectures are all interesting avenues for future work that are currently quite costly due to the lack of an optimized hardware implementation.

Please note that this is merely a technical limitation, not a methodical one as the mLSTM is heavily parallelizable. However, implementing fast compute kernels in CUDA (NVIDIA et al., 2020) or Triton (Tillet et al., 2019) is highly non-trivial as it requires expert hardware architecture knowledge, advanced implementation skills and potentially multiple development cycles to iron out numerical inaccuracies or instabilities.

However, the results of recent linear attention mechanisms show impressive FLOPS utilization (e.g., Yang et al. (2024)). As the mLSTM can be parallelized with similar techniques it is only a matter of time that the mLSTM achieves a similar FLOPS utilization, which will make the mLSTM faster than transformers once an efficient hardware implementation is available.

Additionally, we made a significant effort to make our architecture as efficient as possible, using the tools that are currently available to us. Notably, our architecture is already much faster (up to 70%) than Vim (Zhu et al., 2024) despite Vim using a custom CUDA kernel, as shown in Appendix A.1. For reference, in language modeling, Mamba is roughly on-par with transformers in terms of speed and 4x faster than than the xLSTM (as mentioned in Beck et al. (2024)), again, due to the current lack of efficient hardware implementation of the mLSTM. These considerations further underline the potential of our simple and efficient design for vision applications.

## 6 RELATED WORK

**Generic Vision Backbones.** The inductive bias of CNNs (Fukushima, 1980; LeCun et al., 1998) has demonstrated ground-breaking advancements in computer vision (Krizhevsky et al., 2012) in the early deep learning days. Features of CNNs have been found to learn generic visual features that can be used for a variety of tasks (Donahue et al., 2014). Subsequently, countless works improved various aspects such as architectures (Szegedy et al., 2015; He et al., 2016; Huang et al., 2017; Tan & Le, 2019; Liu et al., 2022) or pre-training strategy (Doersch et al., 2015; Noroozi & Favaro, 2016; Zhang et al., 2016; Gidaris et al., 2018; Chen et al., 2020b; Grill et al., 2020).

**Sequence Models in Vision.** The introduction of transformers (Vaswani et al., 2017) demonstrated exceptional scalability in language processing, which motivated the vision community to explore transformers also in computer vision (Chen et al., 2020a; Cordonnier et al., 2020) but was applied on pixels or small patches which inhibited large costs due to the quadratic complexity of self-attention. This restriction was alleviated by the seminal work Vision Transformers (ViTs) (Dosovitskiy et al., 2021) by using larger patches to aggregate local information and reduce training costs. Similar to CNNs, lots of work improved on the ViT architecture by refining training procedures (Touvron et al., 2021a;b; 2022b; Caron et al., 2021; Bao et al., 2022; Xie et al., 2022; He et al., 2022). The recent advancement of autoregressive models in language processing (Gu & Dao, 2023; Peng et al., 2023) has also gathered interest in the vision community (Zhu et al., 2024; Duan et al., 2024) due to the linear scaling property which allows applications to high-resolution tasks such as medical imaging (Ma et al., 2024) or video understanding (Li et al., 2024).

## 7 CONCLUSION

Motivated by the success of xLSTM in language modeling, we introduced ViL, an adaption of the xLSTM architecture to vision tasks. ViL processes a sequence of patch tokens in alternating fashion. Odd blocks process image patches row-wise from top left to bottom right and even blocks go row-wise from bottom right to top left. Our new architecture outperforms SSM-based vision architectures, other autoregressive vision architectures and also optimized ViT models on ImageNet-1K classification, VTAB-1K transfer classification and ADE20K semantic segmentation. Remarkably, ViL is able to outperform ViT training pipelines, which are the result of years of hyperparameter tuning and transformer improvements.

In the future, we see potential in applying ViL when high-resolution images are needed for optimal performance, such as semantic segmentation or medical imaging. In these settings, transformers suffer from high computational costs due to the quadratic complexity of self-attention, where the linear complexity of ViL allows compute efficient processing of long sequences. Additionally, improving pre-training schemes (e.g., via self-supervised learning), exploring better hyperparameter settings or investigating hierarchical architectures are promising future directions that could improve ViL even further.

## ACKNOWLEDGMENTS

We acknowledge EuroHPC Joint Undertaking for awarding us access to Karolina at IT4Innovations, Czech Republic, MeluXina at LuxProvide, Luxembourg, Leonardo at CINECA, Italy and LUMI at CSC, Finland.

The ELLIS Unit Linz, the LIT AI Lab, the Institute for Machine Learning, are supported by the Federal State Upper Austria. We thank the projects Medical Cognitive Computing Center (MC3), INCONTROL-RL (FFG-881064), PRIMAL (FFG-873979), S3AI (FFG-872172), DL for GranularFlow (FFG-871302), EPILEPSIA (FFG-892171), AIRI FG 9-N (FWF-36284, FWF-36235), AI4GreenHeatingGrids (FFG- 899943), INTEGRATE (FFG-892418), ELISE (H2020-ICT-2019-3 ID: 951847), Stars4Waters (HORIZON-CL6-2021-CLIMATE-01-01). We thank Audi.JKU Deep Learning Center, TGW LOGISTICS GROUP GMBH, Silicon Austria Labs (SAL), FILL Gesellschaft mbH, Anyline GmbH, Google, ZF Friedrichshafen AG, Robert Bosch GmbH, UCB Biopharma SRL, Merck Healthcare KGaA, Verbund AG, GLS (Univ. Waterloo), Software Competence Center Hagenberg GmbH, Borealis AG, TÜV Austria, Frauscher Sensonic, TRUMPF and the NVIDIA Corporation.

## ETHICS STATEMENT

Our work proposes a novel architecture that shows improved FLOPS efficiency compared to previous methods which could greatly reduce energy consumption and carbon emission for future training of large-scale vision or multi-modal models.

We proposes a foundational change in the form of a novel vision architecture, which can be used for a broad set of applications, inheriting their potential benefits, such as enhancing diagnostic accuracy in healthcare, and challenges, such as the risk of improving generative vision models to generate better deepfakes to propagate misinformation.

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

# A    EXTENDED RESULTS

## A.1    RUNTIME COMPARISON OF VIL VS VIM

We compare the runtime to train ViL and Vim (Zhu et al., 2024) for 10 ImageNet-1K epochs in Table 6. We follow the scaling procedure of ViTs, using 192 (T), 384 (S), 768 (B), 1024 (L) as hidden dimension where the (L)arge scale doubles the number of blocks.

Table 6: Runtime comparisons between Vim (Zhu et al., 2024) and ViL. ViL is up to 69% faster despite the current lack of a optimized hardware implementation. As mLSTM (and ViL) can be parallelized analogous to FlashAttention (Dao et al., 2022; Dao, 2023) via custom hardware optimizations, ViL will become even faster in the future. Runtimes denote the training time for 10 ImageNet-1K epochs and are extrapolated from short benchmark runs on a single A100-80GB-PCIe using float16 precision and 224x224 images.

| Model | Optimization | (T)iny | (S)mall | (B)ase | (L)arge |
|---|---|---|---|---|---|
| Vim (Zhu et al., 2024) | custom CUDA kernel | 7.3h | 14.0h | 28.2h | 76.4h |
| ViL | `torch.compile` | 5.0h | 8.7h | 16.6h | 45.1h |
| Speedup of ViL compared to Vim | | 45% | 61% | 69% | 69% |

## A.2    IMPACT OF LONGER TRAINING

We investigate the impact of training for a longer duration in Table 7.

Table 7: Performance comparison of tiny models trained for 400 and 800 epochs. ADE20K mIoU uses single-scale evaluation. All settings follow the ones used in the main paper.

| Model | Epochs | IN-1K ACC | VTAB-1K | ADE20K mIoU |
|---|---|---|---|---|
| DeiT-III-T | 400 | 75.6 | 67.0 | 39.1 |
| DeiT-III-T | 800 | 76.2 | 67.1 | 39.8 |
| ViL-T | 400 | 77.2 | 67.8 | 40.9 |
| ViL-T | 800 | 78.3 | 68.3 | 41.2 |

## A.3    VTAB-1K INDIVIDUAL DATASET RESULTS

Table 8 presents accuracies for each individual dataset of the VTAB-1K benchmark.

Table 8: Results on all datasets of the VTAB-1K (Zhai et al., 2019) benchmark.

| | Natural | | | | | | | Specialized | | | | Structured | | | | | | | |
|---|---|---|---|---|---|---|---|---|---|---|---|---|---|---|---|---|---|---|---|
| | Cifar100 | Caltech101 | DTD | Flower102 | Pets | SVHN | Sun397 | Camelyon | EuroSAT | Resisc45 | Retinopathy | Clevr-Count | Clevr-Dist | DMLab | KITTI-Dist | dSpr-Loc | dSpr-Ori | sNORB-Azim | sNORB-Ele |
| DeiT-T | 47.7 | 86.4 | 63.7 | 85.6 | 87.0 | 78.4 | 35.3 | 83.0 | 93.4 | 80.9 | 70.7 | 71.7 | 60.3 | 43.1 | 78.5 | 67.9 | 41.6 | 30.6 | 32.7 |
| DeiT-III-T | 52.3 | 90.1 | 62.7 | 88.8 | 87.5 | 83.7 | 37.9 | 83.2 | 93.1 | 81.1 | 72.9 | 76.6 | 60.8 | 44.9 | 79.1 | 67.5 | 48.1 | 31.0 | 33.3 |
| Vim-T | 46.7 | 86.3 | 60.7 | 84.0 | 88.8 | 76.1 | 33.7 | 82.2 | 92.9 | 75.2 | 72.6 | 59.8 | 49.9 | 39.3 | 78.2 | 51.2 | 43.9 | 26.9 | 27.2 |
| ViL-T | 54.2 | 90.2 | 67.4 | 90.7 | 89.9 | 81.6 | 41.1 | 83.4 | 94.2 | 82.7 | 73.1 | 80.7 | 61.8 | 49.4 | 81.3 | 57.8 | 51.8 | 31.4 | 34.8 |
| DeiT-S | 57.0 | 88.9 | 68.2 | 90.9 | 90.8 | 75.4 | 42.1 | 83.3 | 94.0 | 83.8 | 74.0 | 74.6 | 58.3 | 45.6 | 78.2 | 61.9 | 47.9 | 27.1 | 31.9 |
| DeiT-III-S | 58.8 | 88.6 | 67.5 | 90.9 | 91.7 | 84.4 | 43.3 | 84.4 | 92.6 | 82.5 | 73.5 | 76.5 | 57.9 | 46.2 | 78.9 | 58.3 | 49.7 | 23.7 | 27.5 |
| Vim-S | 53.0 | 87.2 | 64.1 | 86.8 | 90.3 | 65.8 | 39.7 | 82.4 | 93.4 | 78.0 | 73.1 | 63.1 | 53.2 | 42.3 | 78.2 | 54.1 | 47.6 | 27.1 | 29.3 |
| ViL-S | 61.4 | 89.6 | 69.2 | 92.8 | 91.7 | 78.7 | 43.8 | 85.5 | 93.9 | 84.4 | 73.5 | 84.0 | 63.4 | 51.3 | 83.3 | 61.0 | 55.4 | 32.4 | 35.5 |
| DeiT-B | 61.8 | 89.8 | 67.5 | 93.7 | 92.6 | 84.4 | 45.6 | 85.3 | 95.1 | 86.3 | 74.2 | 77.7 | 59.9 | 47.2 | 81.7 | 61.7 | 51.4 | 30.0 | 36.2 |
| DeiT-III-B | 62.9 | 89.6 | 69.6 | 93.7 | 93.2 | 87.0 | 47.1 | 85.8 | 94.1 | 85.6 | 73.7 | 80.5 | 61.4 | 48.4 | 80.9 | 64.4 | 55.1 | 30.2 | 31.8 |
| ViL-B | 64.3 | 90.0 | 71.1 | 93.4 | 91.4 | 79.6 | 46.6 | 84.7 | 94.3 | 85.4 | 74.4 | 83.7 | 62.1 | 52.7 | 81.0 | 63.1 | 57.6 | 32.6 | 39.9 |

## A.4 Robustness and Domain Generalization

Table 9 presents robustness and OOD evaluations of ImageNet-1K pre-trained classifiers.

Table 9: Robustness and OOD evaluations on ImageNet-C(orruption) (Hendrycks & Dietterich, 2019), ImageNet-A(dversarial) (Hendrycks et al., 2021b), ImageNet-R(endition) (Hendrycks et al., 2021a) and ImageNet-Sketch (Wang et al., 2019).. For ImageNet-C, we report the mean corruption error (Hendrycks & Dietterich, 2019) with AlexNet (Krizhevsky et al., 2012) as baseline.

| Model | IN-C ($\downarrow$) | IN-A ($\uparrow$) | IN-R ($\uparrow$) | Sketch ($\uparrow$) | Validation ($\uparrow$) |
|---|---|---|---|---|---|
| DeiT-T | 69.7 | 7.6 | 32.7 | 19.9 | 72.2 |
| DeiT-III-T | 65.0 | 11.7 | 39.4 | 27.4 | 76.2 |
| Vim-T | 61.8 | 9.6 | 38.8 | 26.9 | 76.1 |
| ViL-T | **59.6** | **15.2** | **42.2** | **30.0** | **78.3** |
| DeiT-S | 54.4 | 19.6 | 41.9 | 29.1 | 79.8 |
| DeiT-III-S | **50.1** | 23.2 | 46.6 | **35.4** | 81.4 |
| Vim-S | 51.5 | 19.7 | 44.8 | 32.5 | 80.5 |
| ViL-S | 50.6 | **23.8** | **47.9** | 35.2 | **81.5** |
| DeiT-B | 48.6 | 27.9 | 44.6 | 32.0 | 81.8 |
| DeiT-III-B | **42.7** | **36.5** | **54.1** | **41.1** | **83.8** |
| ViL-B | 45.3 | 30.9 | 51.9 | 39.0 | 82.4 |

# B Implementation Details

## B.1 Hardware

We train models on servers with either 8xA100 or 4xA100 nodes.

We estimate the total number of A100 GPU-hours used for this project to be 38K hours. This estimate includes initial exploration, method development, analysis and evaluations.

## B.2 FLOPS Calculation

We use the `fvcore`[1] library to count FLOPS and report FLOPS of the mLSTM chunkwise form as described in Section 2.2. For the parallel parts, we report FLOPS for a complexity of $\mathcal{O}\big(\big(\frac{S}{2}+1\big)Sd\big)$ because the upper triangular entries of the $\mathbf{QK}$ matrix do not need to be calculated due to the causal structure. We justify this by the fact that FlashAttention-2 (Dao, 2023) is approximately 1.7x faster with a causal mask than without. Therefore, an optimized hardware implementation of the mLSTM could also omit the calculation of the upper triangular part of $\mathbf{QK}$.

As Vim (Zhu et al., 2024) does not report FLOPS and their model makes use of CUDA kernels (which are not counted as FLOPS by `fvcore`), we replace all calls to CUDA kernels with their reference PyTorch implementation and count the FLOPS with `fvcore`.

For the total pre-training compute in Figure 3, we consider an efficient implementation of stochastic depth (Huang et al., 2016; Touvron et al., 2023) which omits the calculation of a dropped block instead of masking it. Therefore, we change the implementation of ViT (Dosovitskiy et al., 2021) to use our efficient stochastic depth implementation. Vim does not use stochastic depth for training as they only train tiny and small models.

---

[1]https://github.com/facebookresearch/fvcore

## B.3 ViL Hyperparameters

Table 10 shows detailed hyperparameters used to train ViL models.

Table 10: Hyperparameters for training ViL on ImageNet-1K, inspired by DeiT-III (Touvron et al., 2022b). We follow the best setting from DeiT-III (Touvron et al., 2022b) and pre-train on 192 resolution followed by a short fine-tuning on 224 resolution (indicated by $\rightarrow$).

| Parameter | Value |
|---|---|
| Epochs | 800 (T), 400 (S/B) $\rightarrow$ 20 (T, S), 5 (B) |
| Batch size | 2048 $\rightarrow$ 1024 |
| Model | |
|    Patch size | 16x16 |
|    Latent dimension | 192 (T), 384 (S), 768 (B) |
|    Depth | 24 |
|    Pooling | Bilateral Concat |
| Stochastic depth | |
|    Peak rate | 0 (T), 0.05 (S), 0.2 (B) |
|    Layer-wise Decay | ✗ |
| Optimizer | AdamW |
|    Base Learning rate | 1e-3 $\rightarrow$ 1e-5 |
|    Linear LR Scaling Divisor | 1024 |
|    Weight decay | 0.05 |
|    Momentum | $\beta_1 = 0.9, \beta_2 = 0.999$ |
|    Gradient Norm Clip | 1.0 |
| Precision | mixed `bfloat16` |
|    Backend | `torch.autocast` |
| Learning rate schedule | cosine decay |
|    Warmup schedule | linear |
|    Warmup epochs | 5 $\rightarrow$ 5 (T, S), 1 (B) |
|    End LR | 1e-6 |
| Label smoothing | ✗ |
| Train Data Augmentation | |
|    RandomResizedCrop | 192 $\rightarrow$ 224 |
|      Scale | [0.08, 1.0] |
|      Interpolation | bicubic |
|    RandomHorizontalFlip | $p = 0.5$ |
|    3-Augment | |
|      Gaussian Blur $\sigma$ | [0.1, 2.0] |
|      ColorJitter | [0.3, 0.3, 0.3, 0.0] |
|    Normalize | ImageNet-1K statistics |
|    Mixup $\alpha$ | 0.8 |
|    Cutmix $\alpha$ | 1.0 |
| Test Data Augmentation | |
|    Resize | 192 $\rightarrow$ 224 |
|      Interpolation | bicubic |
|    CenterCrop | 192 $\rightarrow$ 224 |
|    Normalize | ImageNet-1K statistics |

## B.4 FINE-TUNING ON VTAB-1K

For fine-tuning models on VTAB-1K we provide the hyperparameters in Table 11. We search for the best learning rate for each dataset by fine-tuning the model 25 times (5 learning rates with 5 seeds each) on the 800 training samples and evaluating them on the 200 validation samples. With the best learning rate, we then train each model 5 times on concatenation of training and validation split, evaluate on the test split and report the average accuracy.

Table 11: Hyperparameters for fine-tuning on VTAB-1K. *For Vim and ViL we group two consecutive blocks for the layer-wise lr decay similar to how ViT considers a pair of attention and MLP block as a single "layer" for the decay.

| Parameter | Value |
|---|---|
| Epochs | 50 |
| Batch size | 64 |
| Seeds | 5 |
| Optimizer | AdamW |
|    Learning rate | [1e-3, 7.5e-4, 5.0e-4, 2.5e-4, 1.0e-4] |
|    Layer-wise lr deca | 0.65* |
|    Weight decay | 0.05 |
|    Momentum | $\beta_1 = 0.9, \beta_2 = 0.999$ |
| Learning rate schedule | linear warmup $\rightarrow$ cosine decay |
|    Warmup epochs | 5 |
| Precision | mixed `bfloat16` |
|    Backend | `torch.autocast` |
| Data Augmentation | |
|   `Resize` | |
|     `interpolation` | bicubic |
|     `size` | 224x224 |
|   `Normalize` | ImageNet-1K statistics |

### B.5    ADE20K Semantic Segmentation Fine-tuning

We fine-tune models on ADE20K (Zhou et al., 2019) using an UperNet (Xiao et al., 2018) head. We follow common practices and fine-tune on 512x512 resolution, where we interpolate the absolute positional embedding from 224x224 to 512x512. For ViTs, we add relative position biases to the attention layers (initialized to 0) (He et al., 2022). Table 12 lists detailed hyperparameters.

Table 12: Hyperparameters for fine-tuning on VTAB-1K. *For ViL we group two consecutive blocks into one similar to how a ViT block consists of a pair of attention and MLP block.

| Parameter | Value |
|---|---|
| Updates | 160K |
| Batch size | 16 |
| UperNet | |
|   Auxiliary | |
|     Weight | 0.4 |
|     Input Block | 8* |
|     Dimension | 192 (T), 384 (S, B) |
|   Decoder | |
|     Weight | 1.0 |
|     Input Blocks | [4, 6, 8, 12]* |
|     Dimension | 192 (T), 384 (S, B) |
| Stochastic depth | |
|   Peak rate | 0 (T), 0.05 (S), 0.1 (B) |
|   Layer-wise Decay | ✓ |
| Optimizer | AdamW |
|   Learning rate | 5e-4 |
|   Linear LR Scaling Divisor | 16 |
|   Layer-wise lr decay | 0.65* |
|   Weight decay | 0.05 |
|   Momentum | $\beta_1 = 0.9, \beta_2 = 0.999$ |
| Learning rate schedule | linear warmup $\rightarrow$ cosine decay |
|   Warmup updates | 1500 |
| Precision | mixed `float16` |
|   Backend | `torch.autocast` |
| Train Data Augmentation | |
|   `RandomResize` | |
|     `interpolation` | bicubic |
|   `RandomCrop` | |
|     `size` | 512x512 |
|   `RandomHorizontalFlip` | |
|   `ColorJitter` | 0.5 |
|     `brightness` | 0.5 |
|     `contrast` | 0.5 |
|     `saturation` | 0.5 |
|     `hue` | 0.25 |
|   `Normalize` | ImageNet-1K statistics |
| Evaluation | |
|   `Stride` | 341 |
|   `Multi-scale` | |
|     `scale factors` | [0.75, 1.0, 1.25, 1.5, 1.75] |
|     `flip` | [True, False] |

## B.6 DEIT-III REIMPLEMENTATION HYPERPARAMETERS

Table 10 shows detailed hyperparameters used to train DeiT-III-T (reimpl.) from Table 1. Our reimplementation easily outperforms older baselines like DeiT-II-T (+2.7% ImageNet-1K accuracy) and is approximately even with the original on ADE20K (40.1 vs 39.8 on mIoU single-scale, 41.8 vs 42.2 mIoU multi-scale).

Table 13: Hyperparameters for training our reimplementation of DeiT-III-T (Touvron et al., 2022b) on ImageNet-1K. The most significant change is that we reduce the learning rate from 3e-3 to 1e-3 as we found this to greatly improve performance. We make minor changes to the protocol such as using AdamW or no gradient clipping as models were stable without it. We follow the best setting from DeiT-III (Touvron et al., 2022b) and pre-train on 192 resolution followed by a short fine-tuning on 224 resolution (indicated by $\rightarrow$).

| Parameter | Value |
|---|---|
| Epochs | $800 \rightarrow 20$ |
| Batch size | $2048 \rightarrow 1024$ |
| Model | |
|     Patch size | 16x16 |
|     Latent dimension | 192 |
|     Depth | 12 |
|     Pooling | [CLS] |
| Stochastic depth | ✗ |
| Layerscale | 1e-4 |
| Optimizer | AdamW |
|     Base Learning rate | $1e\text{-}3 \rightarrow 1e\text{-}5$ |
|     Linear LR Scaling Divisor | 1024 |
|     Weight decay | 0.05 |
|     Momentum | $\beta_1 = 0.9, \beta_2 = 0.999$ |
|     Gradient Norm Clip | ✗ |
| Precision | mixed `bfloat16` |
|     Backend | `torch.autocast` |
| Learning rate schedule | cosine decay |
|     Warmup schedule | linear |
|     Warmup epochs | 5 |
|     End LR | 1e-6 |
| Label smoothing | ✗ |
| Train Data Augmentation | |
|     RandomResizedCrop | $192 \rightarrow 224$ |
|       Scale | [0.08, 1.0] |
|       Interpolation | bicubic |
|     RandomHorizontalFlip | $p = 0.5$ |
|     3-Augment | |
|       Gaussian Blur $\sigma$ | [0.1, 2.0] |
|       ColorJitter | [0.3, 0.3, 0.3, 0.0] |
|     Normalize | ImageNet-1K statistics |
|     Mixup $\alpha$ | 0.8 |
|     Cutmix $\alpha$ | 1.0 |
| Test Data Augmentation | |
|     Resize | $192 \rightarrow 224$ |
|       Interpolation | bicubic |
|     CenterCrop | $192 \rightarrow 224$ |
|     Normalize | ImageNet-1K statistics |

