# OpenReview forum: "Vision-LSTM: xLSTM as Generic Vision Backbone"
_ICLR.cc/2025/Conference — ICLR 2025 Poster_

### Official Review · Reviewer_DdR3 · 2024-10-26

**Soundness:** 3
**Presentation:** 2
**Contribution:** 2
**Rating:** 5
**Confidence:** 4

**Summary:**

This paper presents a new vision backbone based on Extended Long Short-Term Memory (xLSTM), which is first proposed in NLP domain. The model ViL consists of a linear projection, a list of mLSTM blocks, and a prediction head. In each mLSTM block, the image feature patches go through a recurrent process with modified LSTM nodes. The model can be configured using a chunkwise mode to enable efficient parallel processing. The experiments show the proposed model achieve competitive performance on several vision tasks: imageNet classification, semantic segmentation, and transfer learning.

**Strengths:**

-	The proposed framework is clean and straightforward.
-	The model achieves competitive performance on classification, segmentation, and transfer learning comparing to existing models vision transformer, vision-mamba and ConvNeXt.
-	On segmentation tasks, it is able to outperform existing methods with lower FLOPs, due to the efficiency of the recurrent processing on high image resolutions.

**Weaknesses:**

-	The overall novelty of the work is limited. The framework is directly adapted from xLSTM in NLP. The benefit on parallel inference is also directly inherited from the xLSTM framework. The reviewer would like to see some modifications to the original framework to make the model more suitable for the image-domain tasks, such as incorporating image priors or efficient message passing.
-	In figure 3, it seems the performance of ViL saturates faster than ViT models. On ImageNet, it is worse than DeiT-III on large-size settings. Not sure the scalability of this work if the model size gets even larger. In Table 1, ViL-B is also not as good as Mamba-B. More results on larger models is referred to make the experiments stronger.
-	There are no latency numbers on CPU/GPUs for ViL. Such numbers are critical for new vision backbones.

**Questions:**

What is the memory consumption / latency of the xLSTM compared to vision mamba? Does it have any advantage?

---

> ### Author Response · Authors · 2024-11-18
>
> We appreciate your thorough review and are happy to see that our framework was considered clean and easy to understand. We address your concerns below.
>
> **Integration of image priors**
>
> While our work undeniably leverages the xLSTM backbone and inherits its advantages and disadvantages alike, we do make a considerable amount of changes to make the model suitable for image tasks. For example Table 4 (a) shows that the additional backward traversal greatly improves performance as it efficiently incorporates the non-causal structure of images into the model.
>
>
> **Classification performance vs DeiT-III**
>
> As also discussed in the general response of the rebuttal, DeiT-III is the result of multiple iterations of large scale hyperparameter tuning of transformers for image classification on ImageNet. Contrary, our work is the first one to apply xLSTM to this kind of task and, consequently, the exploration of the vast hyperparameter space is nowhere near the same level. While we do take inspiration from insights of the DeiT series, transformers and xLSTM are two fundamentally different models where it is highly unlikely that both architectures have the same optimal hyperparameters.
>
> The performance of ViL-B (82.4\%) is between DeiT-B (81.8\%) and DeiT-II-B (82.7\%), which we would argue reflects the level of hyperparameter tuning quite accurately. Notably, we do not consider the original ViT (77.9\%) [1] as a sensible baseline due to the advancements made since then. Therefore, DeiT-II is the result of the third iteration of ViT hyperparameter tuning and improvements.
>
>
> Additionally, the strong ImageNet-1K pre-training performance of DeiT-III-B does not translate to segmentation or transfer learning results which suggests that the DeiT-III training protocol is highly specialized, maybe even overspecialized, to natural image classification. In particular, DeiT-III-B outperforms ViL on ImageNet-1K and the natural category of VTAB-1K, showing good results on natural image classification. However, ViL-B shows excellent results on semantic segmentation and the structural category of VTAB-1K.
>
> We want to provide the reader with the full picture by comparing against the current state-of-the-art baselines. We aim to strengthen understanding of vision models by highlighting their respective strengths and weaknesses. At the moment, larger models of highly optimized ViT pipelines beat ViL in natural image classification, but at the same time ViL beats them in structured tasks like semantic segmentation. This highlights the strengths and weaknesses of the respective methods, which we believe is a valuable insight. In contrast, other vision sequence models (e.g, [2, 3, 4]) simply omit DeiT-II/DeiT-III baseline models to make their results look better, concealing the fact that sequence models are not at the performance of optimized transformers in natural image classification on larger scales yet. We refrain from this practice and instead opt for transparency by reporting state-of-the-art models as baselines instead of outdated ones.
>
>
>
> [1] Dosovitsky 2020, "An Image is Worth 16x16 Words: Transformers for Image Recognition at Scale" https://arxiv.org/abs/2010.11929
>
> [2] Zhu 2024, "Vision Mamba: Efficient Visual Representation Learning with Bidirectional State Space Model" https://arxiv.org/abs/2401.09417
>
> [3] Wang 2024, "Mamba-R: Vision Mamba ALSO Needs Registers" https://arxiv.org/abs/2405.14858
>
> [4] Duan 2024, "Vision-RWKV: Efficient and Scalable Visual Perception with RWKV-Like Architectures" https://arxiv.org/abs/2403.02308
>
>
> **Classification performance vs Mamba-R**
>
> Indeed, Mamba-R shows good ImageNet-1K classification with its 99M parameter model. However, it also has 10M more parameters and 2.7GFLOPS more than ViL-B. We would argue that the 0.5\% performance difference can easily stem from the increased model size of Mamba-R-B. Additionally, ViL-B easily outperforms Mamba-R-B on semantic segmentation despite using less parameters and FLOPS.
>
> In similar vein as to why we report DeiT-II and DeiT-III, despite them being an highly optimized architecture, we want to show a comprehensive comparison to the reader. Mamba-R is an advancement of Vision-Mamba that introduces additional register tokens to avoid high-norm tokens. Consequently, it is not a completely fair comparison to  ViL as Mamba-R is already an optimized architecture. Similar analysis and insights into ViL could bring similar insights and motivate future design choices of ViL to improve performance even further.

---

> ### Author Response · Authors · 2024-11-18
>
> **Latency numbers for ViL**
>
> We report training times in Appendix A.1 where we show that ViL is already up to 70\% faster than Vision-Mamba despite the lack of optimized hardware optimization.
>
> We agree that latency is a crucial aspect of new backbones. However, the current lack of optimized hardware implementation for the mLSTM makes FLOPS the most accurate representation of the runtime of our model that we currently have. That is because the mLSTM currently has significantly higher latency than the attention mechanism due to lack of optimized hardware implementations, where transformers have undergone multiple optimization iterations.
>
> Additionally, as extensively discussed in the general response, the results of recent linear attention mechanisms show impressive FLOPS utilization (e.g., [1]). As the mLSTM can be parallelized with similar techniques it is only a matter of time that the mLSTM achieves a similar FLOPS utilization, which will make the mLSTM faster than transformers once an efficient hardware implementation (on which we are actively working on) is available.
>
> We expanded Section 5 to include this discussion.
>
>
> [1] Yang 2023, "Gated Linear Attention Transformers with Hardware-Efficient Training" https://arxiv.org/abs/2312.06635
>
> **Memory consumption vs Vision-Mamba**
>
> Optimized hardware implementation not only drastically reduce runtime, but also reduce memory consumption by a lot. Therefore, Vision-Mamba, while up to 70\% slower than Vision-LSTM, currently has smaller memory footprint. However, FlashAttention [2] or recent optimized linear attention implementations (e.g., [3]) also reduce the required memory by a huge amount, which means that also an optimized mLSTM implementation will have a much smaller memory consumption.
>
> As the internal cell state $C$ of the mLSTM has constant size, it has constant theoretical memory complexity (same as Mamba). However, due to our efficient alternating block design, ViL might even achieve lower memory consumption than Vim as Vim uses two Mamba state-space-models per block, whereas ViL uses only a single mLSTM per block.
> Therefore, we could even imagine that ViL will be more memory efficient than Vim, once an efficient hardware implementation is available.
>
>
> [2] Dao NeurIPS 2022, "FlashAttention: Fast and Memory-Efficient Exact Attention with IO-Awareness" https://arxiv.org/abs/2205.14135
>
>
> [3] Yang 2023, "Gated Linear Attention Transformers with Hardware-Efficient Training" https://arxiv.org/abs/2312.06635

---

> > ### Comment · Reviewer_DdR3 · 2024-11-27
> >
> > Thanks a lot for the detailed response. While it addresses some of the concerns (latency and memory usages, potential optimizations), I still have major concerns on the novelty of the work due to lack of technical differences from existing x-LSTM framework. This is also reflected by reviewers xJrX.

---

### Official Review · Reviewer_JQin · 2024-10-27

**Soundness:** 3
**Presentation:** 3
**Contribution:** 3
**Rating:** 5
**Confidence:** 3

**Summary:**

This paper introduces Vision-LSTM (ViL), a novel generic backbone for computer vision tasks that adapts the recently proposed xLSTM architecture to vision. ViL processes images by splitting them into patches and processing sequences of patch tokens using alternating xLSTM blocks. Odd blocks process the sequence from top-left to bottom-right, while even blocks process it from bottom-right to top-left. The key advantage of ViL is its linear computational and memory complexity with respect to sequence length, which makes it more efficient than Transformers for high-resolution images. The authors conduct experiments on ImageNet-1K classification, ADE20K semantic segmentation, and VTAB-1K transfer learning tasks, showing that ViL achieves competitive or superior performance compared to Vision Transformers (ViT), Vision Mamba (Vim), and other recent architectures. They also provide ablation studies on architectural design choices and discuss limitations and future work.

**Strengths:**

Originality: The paper presents a novel adaptation of the xLSTM architecture to computer vision tasks, which is a creative application of recent advancements in sequence modeling to vision.

Quality: The experimental evaluation is thorough, including comparisons with strong baselines on standard benchmarks (ImageNet-1K, ADE20K, VTAB-1K). The authors also perform ablation studies to justify architectural choices.

Significance: The proposed ViL architecture offers linear computational and memory complexity with respect to sequence length, addressing a key limitation of Transformers (quadratic complexity) in high-resolution image tasks. This has potential implications for scaling vision models to higher resolutions.

Clarity: The paper is well-written and clearly explains the methodology, experiments, and results. The figures and tables are informative and enhance understanding.

**Weaknesses:**

Practical Implementation: The lack of an optimized hardware implementation of the mLSTM limits the practical runtime performance of ViL compared to ViTs, which benefit from highly optimized libraries. This may hinder immediate adoption of ViL in real-world applications.

Scope of Experiments: While the experiments are comprehensive for the given datasets, the evaluation is limited to ImageNet-1K and related benchmarks. Larger-scale pre-training on datasets like ImageNet-21K or JFT-300M could strengthen the claims and demonstrate scalability.

Limited Exploration of Extensions: The paper mentions potential future work such as self-supervised pre-training and hierarchical architectures but does not explore these avenues. Including preliminary results or discussions on these topics could enhance the paper's contribution.

Technical Limitations Affecting Design Choices: Some architectural design choices, such as limiting traversal directions due to technical constraints (lack of optimized implementations), suggest that the current version of ViL may not be fully optimized from a methodological standpoint.

**Questions:**

Could the authors compare against transformers under wall clock time instead of FLOPs?
Could the authors provide more details on the potential for optimized hardware implementations of the mLSTM, and how this might impact practical runtimes compared to ViTs? Are there any ongoing efforts in this direction?
Have the authors considered applying ViL to larger-scale datasets or tasks that particularly benefit from high-resolution inputs, such as medical imaging or video understanding? How do they anticipate ViL would perform in these settings?

---

> ### Author Response · Authors · 2024-11-18
>
> Thank you for your review and helpful comments to help us improve the paper. We address your points individually.
>
>
> **Lack of optimized hardware implementation**
>
> While we recognize that the lack of optimized hardware implementation is a limiting factor of ViL (as discussed in the limitation section and the general response of the rebuttal), it is important to consider that the xLSTM is a new architecture whereas transformers have been optimized for over 7 years by now. We are actively working on the development of an optimized hardware implementation to foster adaptation of ViL in real-world applications. However, this is a highly non-trivial project that requires expert domain knowledge in specialized programming languages as well as a deep understanding of numerical precisions, hardware layouts, parallel processing, compute to memory interactions and many more areas. This is underlined by the fact that optimized hardware implementations are often standalone papers in major conferences (e.g. [1, 2]).
>
>
> [1] Dao NeurIPS 2022, "FlashAttention: Fast and Memory-Efficient Exact Attention with IO-Awareness" https://arxiv.org/abs/2205.14135
>
> [2] Dao ICLR 2024, "FlashAttention-2: Faster Attention with Better Parallelism and Work Partitioning" https://arxiv.org/abs/2307.08691
>
> **Comparison under wall clock time**
>
> We agree that FLOPS are not the ideal metric to report, however, it is the most accurate representation of the runtime of our model that we currently have. The mLSTM is still significantly slower than the attention mechanism due to lack of optimized hardware implementations, where transformers have undergone multiple optimization iterations. Nevertheless, we do report runtimes relative to Vision-Mamba in Appendix A.1 where we underline the potential of ViL as efficient vision sequence model as it has up to 70\% lower runtime despite Vision-Mamba having a custom CUDA kernel.
>
> Additionally, as extensively discussed in the general response, the results of recent linear attention mechanisms show impressive FLOPS utilization (e.g., [3]). As the mLSTM can be parallelized with similar techniques it is only a matter of time that the mLSTM achieves a similar FLOPS utilization, which will make the mLSTM faster than transformers once an efficient hardware implementation (on which we are actively working on) is available.
>
> We expanded Section 5 to include this discussion.
>
>
> [3] Yang 2023, "Gated Linear Attention Transformers with Hardware-Efficient Training" https://arxiv.org/abs/2312.06635
>
> **Pre-training on larger datasets**
>
> We agree that pre-training on larger datasets would further strengthen the paper, but want to highlight that not all competing methods pre-train models on ImageNet-21K, which heavily limits comparability. We instead opt for an extensive evaluation of ImageNet-1K pre-trained models which is a standard benchmark that all compared methods report and publish models for. Additionally, pre-training schedules on ImageNet-21K are much longer than the standard ImageNet-1K schedulees, making it extremely expensive to train models on. Please also note that it is impossible for us to pre-train models on JFT-300M as it is a private dataset only available to researchers from Google and DeepMind.
>
> **Application to domains benefiting from high-resolution**
>
> The currently largest resolution that we applied ViL to is the 512x512 from the ADE20K semantic segmentation task, which is somewhat high resolution where ViL already sees a big advantage in terms of FLOPS due to its linear complexity and outperforms competitors, often by quite a large margin (see Table 2). While we envision that ViL can achieve strong performances in other domains such as 3D medical image segmentation, 3D point cloud processing or DNA modeling, we leave exploration thereof to future work.
>
>
> **Further optimization of ViL**
>
> We agree with the reviewer that the current ViL architecture is not fully optimized as the hyperparameter space is extremely large. However, as ViL shows strong performances across various tasks, we believe that the current state provides a nice starting point for adaption to various other domains and further optimization studies, which we consider to be a valuable contribution.

---

### Official Review · Reviewer_wC79 · 2024-11-03

**Soundness:** 2
**Presentation:** 3
**Contribution:** 3
**Rating:** 6
**Confidence:** 4

**Summary:**

This work bringing the advantages of LSTM to computer vision to build a new vision backbone with linear complexity.  Previous vision backbone such as Vision Transformer and  VIsion Mamba have challenges in the computational complexity of processing high-resolution image tasks, so this work extends LSTM by extending the gating and parallelizable matrix memory structure to address long-standing limitations. The proposed backbone Vision LSTM (ViL) is validated in image classification, semantic segementation etc.

**Strengths:**

1.  The new attemption of new linear vision backbone is great.
2.  This work has detailed experimental setup in classification, transfer learning and segmentation. ViL performs well on ImageNet accuracy, ADE20K mIoU and VTAB-1K accuracy.

**Weaknesses:**

1. Because of the good training receipt (data augmentation, optimization method etc.), it is not difficult to get good performance to train a new vision backbone. My main concern is how to validate the scaling law of a new backbone, namely the proposed ViL.

2. The largest model size of ViL is 89M and 115M (ViL-B), so how to validate the performance still can keep spurious with larger model size.

**Questions:**

Please refer to the weaknesses.

---

> ### Author Response · Authors · 2024-11-18
>
> We appreciate your helpful review and are happy to see that our linear vision backbone was well received and the extensive experiments were appreciated. We address your concerns below.
>
> **Scaling to larger model sizes**
>
> As also discussed in the general response and in the response to reviewer VTdU, we agree with the reviewer that one would ideally show performance on even larger models. However, as there is currently no efficient hardware implementation of the mLSTM available, training is still quite slow which makes larger scales expensive. Nevertheless, we would argue that our contributions give valuable insights into vision sequence models that strengthen the understanding for researchers and practitioners, even without training 300M parameter models. Please note that also early works of transformers (e.g., DeiT) or mamba (e.g., Vim) did not train extremely large models. As implementations got better and compute got cheaper, it made training of larger models more accessible which allowed follow-up works also train larger models (e.g., DeiT-III, Mamba-R).
>
> Additionally, given the scaling curves of the xLSTM in language modeling, we see no reason why ViL would not scale to larger scales, as both use the mLSTM as their core component.

---

> > ### Comment · Reviewer_wC79 · 2024-11-26
> > **How about the training and inference latency of ViL?**
> >
> > My other concern is about the training and inference latency of ViL, comparing with ViM and ViT.
> > As LSTM is non-parallel compared with Transformer, inference latency would be a very important problem.

---

> > > ### Author Response · Authors · 2024-11-26
> > >
> > > **Vision-LSTM is fully parallelizable**
> > >
> > > We use a modernized form of the LSTM, namely the mLSTM of the xLSTM [1] family. The mLSTM is an advanced version of the vanilla LSTM that can be formulated in a fully parallelizable way, similar to transformers. In contrast to the vanilla LSTM, it **does not have a recurrence relation**. Vision-LSTM uses only mLSTM blocks, i.e. the whole model is fully parallelizable (like transformers).
> > >
> > > Other LSTM variants like the vanilla LSTM or the sLSTM have recurrence relations and are therefore much slower as their forward pass cannot be fully parallelized. **We do not use any of these components in ViL**.
> > >
> > > **Latency comparison**
> > >
> > > We compare training runtimes in Appendix A.1 where ViL is up to 70% faster than Vision-Mamba despite Vision-Mamba using a custom CUDA kernel. Vision-LSTM is faster than Vision-Mamba because of our efficient design choices in the architecture (e.g., our alternating direction block design). As transformer implementations are highly optimized, Vision-Transformers are still faster than ViL. However, once ViL implementations reach a similar FLOPS throughput as transformers (e.g., with a custom hardware implementation of the mLSTM), ViL will be as fast as ViTs on small resolutions and much faster than ViT on higher resolutions (as shown in Figure 1 right).
> > >
> > > We base this claim on the fact that the mLSTM can be implemented analogous to recent linear attention mechanisms, which achieved impressive FLOPS throughputs (e.g., [2]). Therefore, once we have an optimized mLSTM implementation, runtimes should be highly correlated with FLOPS and, as shown in Figure 1 right, ViL has less FLOPS than Vim and also less FLOPS than ViT, particularly on higher resolutions.
> > >
> > >
> > >
> > > [1] Beck NeurIPS 2024, "xLSTM: Extended Long Short-Term Memory" https://arxiv.org/abs/2405.04517
> > >
> > > [2] Yang 2023, "Gated Linear Attention Transformers with Hardware-Efficient Training" https://arxiv.org/abs/2312.06635

---

> > > > ### Comment · Reviewer_wC79 · 2024-11-27
> > > > **Some of my concerns are addressed.**
> > > >
> > > > The problem of scaling law of ViL is still a problem. Some of my concerns are addresses, so I arise the score to 6.

---

### Official Review · Reviewer_xJrX · 2024-11-03

**Soundness:** 3
**Presentation:** 2
**Contribution:** 3
**Rating:** 6
**Confidence:** 3

**Summary:**

This paper applies the recently proposed language model architecture xLSTM to the field of computer vision, and the proposed backbone is named Vision-LSTM (ViL). The image is firstly separated into patch tokens as in ViT, then ViL applies xLSTM module to image tokens in the bidirectional order since the image is non-causal data. This paper compares ViL with other vision backbones (ViT, DeiT, etc.) in three vision tasks, including classification, semantic segmentaion and transfer learning, where the experiments show that ViL achieves strong performances and also demonstrates a good trade-off between performance and computations.

**Strengths:**

1. The proposed ViL display xLSTM also performs well in visual feature encoding and can be considered a strong candidate for a universal visual backbone.
2. Extensive experiments are conducted to verify the strong performance of ViL on three vision tasks.

**Weaknesses:**

1. The technical contribution is limited: the proposed ViL is a simple adaptation of xLSTM blocks to vision tasks. Although it contains some necessary modifications for processing non-causal image data (bidirectional flip, conv2d, etc.), it is still straightforward.
2. Lack of experiments to prove the main advantages of ViL: compared with transformers, the ViL has linear complexity. But the experiments do not provide enough evidence to show this advantage. For example, the mentioned lack of an optimized hardware implementation could also be a potential and important technical contribution for ViL (a prototype implementation should also be a good contribution). Adding an ablation study to demonstrate the effectiveness of ViL for processing higher-resolution / using larger models should also be a good choice.
3. The writing can be further improved and the paper should be self-contained: Section 2.1 lists lots of equations about mLSTM, but they are not used. Actually, the introduction of mLSTM is also difficult to understand with these equations and limited text. Additionally, the paper length is a little shorter than 10 pages.

**Questions:**

For semantic sementation task, does ViL use feature pyramid and how to implement it?

Please also refer to the weakness section, especially for the second and third points.

---

> ### Author Response · Authors · 2024-11-18
>
> Thank you for your valuable review and suggestions to further improve the clarity of our paper. We are happy that our architecture was found to be a strong candidate for a new universal vision backbone. We respond to your questions below.
>
> **Novelty of ViL**
>
> While we agree with the reviewer that the introduced modifications are not ground-breaking inventions, we would argue that they are not that straight forward. For example, while the Vision Mamba line of work (e.g., [1, 2]) has had great success and lots of people adapted it (as highlighted by the almost 600 citations of Vim within less than a year), they are still using 1D convolutions and non-alternating blocks which make the models significantly slower. We therefore believe that our approach is of value to the community and could bring new ideas also to other model architectures.
>
> Additionally, computer vision is an extremely competitive field where countless works are concerned with studying hyperparameters and optimizing architectures to maximize performance (e.g., [3, 4, 5, 6, 7]). These works often have a large impact on the community despite "little novelty" as narrowing down the vast search space of architectures and hyperparameters advances the understanding of vision models for researchers and practitioners.
>
>
> [3] Touvron 2020, "Training data-efficient image transformers & distillation through attention" https://arxiv.org/abs/2012.12877
>
> [4] Touvron, "DeiT III: Revenge of the ViT" https://arxiv.org/abs/2204.07118
>
> [5] Liu 2022, "A ConvNet for the 2020s" https://arxiv.org/abs/2201.03545
>
> [6] Wightman 2021, "ResNet strikes back: An improved training procedure in timm" https://arxiv.org/abs/2110.00476
>
> [7] Touvron 2022, "Three things everyone should know about Vision Transformers" https://arxiv.org/abs/2203.09795
>
>
> **Optimized hardware implementation as contribution**
>
> As also extensively discussed in the general response, developing an optimized hardware implementation is not something trivial and requires expert domain knowledge in specialized programming languages as well as a deep understanding of numerical precisions, hardware layouts, parallel processing, compute to memory interactions and many more areas. This is underlined by the fact that optimized hardware implementations are often standalone papers (e.g. [8, 9]).
>
> While we agree that the current lack of optimized hardware implementation is a limitation of ViL (as also discussed throughout the paper), it does not change any of the insights from our paper, which we believe are of value to the community to strengthen the understanding of vision sequence models.
>
> [8] Dao NeurIPS 2022, "FlashAttention: Fast and Memory-Efficient Exact Attention with IO-Awareness" https://arxiv.org/abs/2205.14135
>
> [9] Dao ICLR 2024, "FlashAttention-2: Faster Attention with Better Parallelism and Work Partitioning" https://arxiv.org/abs/2307.08691
>
> **Experiments to leverage linear complexity**
>
> We aim to show the potential of ViL for high resolution images on the left side of Figure 1, where we show that ViL has less FLOPS than Vision-Tranformer and Vision-Mamba, particularly on higher resolutions. While we do not consider FLOPS to be the perfect metric for showing this, and runtime would be preferred, a runtime comparison would obviously favor ViTs as transformers were introduced 7 years ago and consequently have highly optimized implementations whereas the mLSTM currently does not have an optimized hardware implementation yet. However, as the mLSTM is fully parallelizable, the mLSTM will outperform transformers also in terms of runtime given a fast hardware implementation, particularly on higher resolutions.
>
>
> **Writing improvements**
>
> As also suggested by the reviewer, we aim to make the paper self-contained and therefore introduce the mLSTM forward pass in the method section. We see how the mLSTM section is a bit hard to follow and therefore restructured it and added additional information to put it better into the context of vision sequence models. The mLSTM fills the role of exchanging information between patches, similar to what the attention does in vision transformers. We also added this missing information to Figure 1.
>
>
> **Semantic segmentation feature pyramids**
>
> We use a standard evaluation pipeline for semantic segmentation which uses an UperNet head that uses a feature pyramid as input. We follow the standard procedure that takes the patch tokens after the 4th, 6th, 8th and final blocks and upsamples the outputs of the 4th and 6th block to a higher resolution while downsampling the final block to smaller resolution. These features are then processed in an UperNet head that uses convolution and pyramid pooling to classify each pixel.

---

> ### Author Response · Authors · 2024-11-18
>
> **Less than 10 pages**
>
> Please note that using less than 10 pages is explicitly encouraged in the ICLR 2025 call for papers: "We encourage authors to be crisp in their writing by submitting papers with 9 pages of main text. We recommend that authors only use the longer page limit in order to include larger and more detailed figures. However, authors are free to use the pages as they wish, as long as they obey the page limits."

---

> ### Comment · Reviewer_xJrX · 2024-12-01
>
> Thanks a lot for the authors' detailed response, some of my concerns have been addressed so I raise my score to 6.
>
> However, I still have reservations regarding the limited technical contributions and the lack of a hardware-aware prototype (e.g., CUDA) to demonstrate the module’s efficiency. Some points were also raised by reviewers DdR3, VTdU, and JQin.

---

### Official Review · Reviewer_VTdU · 2024-11-05

**Soundness:** 3
**Presentation:** 3
**Contribution:** 3
**Rating:** 6
**Confidence:** 5

**Summary:**

The paper introduces VIsion-LSTM, a novel general-purpose vision backbone building on top of xLSTM. The proposed ViL models show good visual understanding performance in various tasks (e.g., image classification, semantic segmentation) and exhibit superior inference speed over Vision Transformer and Vision Mamba.

**Strengths:**

1. It's interesting to try transfering different language models into vision. As Tranformer has shown a very successful adaptation in computer vision and Mamba has recently been introduced into various vision tasks, showing comparable performance, validating the similar effect of LSTM can provide many insights to the community.

2. The performance is good. xLSTM shows competitve results in classification and semantic segmentation.

3. The detailed ablations of architectural design are interesting and can inspire future works.

**Weaknesses:**

1. On the ImageNet-1k classification task, the model seems not to scale well. The ViL-Base underperforms DeiT-III by a large margin. Is this caused by a technical reason (e.g., insufficient hyper-parameter search) or the limitation of LSTM's learning capacity? Can ViL scale to a larger size?

2. In the main tables of the paper, the authors emphasize comparing the models' FLOPs as a measure of speed, which may not be a fair comparison between recurrent models and transformers. Typically, at the same FLOPs or inference speed, recurrent models train significantly slower than transformers. Did the authors provide a direct comparison of speed during the training stage?

3. Some considerable performance improments come from the 2D convolution.

**Questions:**

Is ViL a plain architecture or involving feature downsampling operations? It is not clearly disscussed in the paper.

---

> ### Author Response · Authors · 2024-11-18
>
> Thank you for your profound review and suggestions that helped us to improve our paper. We are glad that our new vision architecture and detailed ablations were found to be valuable. We address your points below.
>
> **Classification performance vs DeiT-III**
>
> DeiT-III is the result of multiple iterations of large scale hyperparameter tuning of transformers for image classification on ImageNet. Contrary, our work is the first one to apply xLSTM to this kind of task and, consequently, the exploration of the vast hyperparameter space is nowhere near the same level. While we do take inspiration from insights of the DeiT series, transformers and xLSTM are two fundamentally different models where it is highly unlikely that both architectures have the same optimal hyperparameters.
>
> The performance of ViL-B (82.4\%) is between DeiT-B (81.8\%) and DeiT-II-B (82.7\%), which we would argue reflects the level of hyperparameter tuning quite accurately. Notably, we do not consider the original ViT (77.9\%) [1] as a sensible baseline due to the advancements made since then. Therefore, DeiT-II is the result of the third iteration of ViT hyperparameter tuning and improvements.
>
>
> Additionally, the strong ImageNet-1K pre-training performance of DeiT-III-B does not translate to segmentation or transfer learning results which suggests that the DeiT-III training protocol is highly specialized, maybe even overspecialized, to natural image classification. In particular, DeiT-III-B outperforms ViL on ImageNet-1K and the natural category of VTAB-1K, showing good results on natural image classification. However, ViL-B shows excellent results on semantic segmentation and the structural category of VTAB-1K, outperforming both DeiT-II and DeiT-III.
>
> We want to provide the reader with the full picture by comparing against the current state-of-the-art baselines. We aim to strengthen understanding of vision models by highlighting their respective strengths and weaknesses. At the moment, larger models of highly optimized ViT pipelines beat ViL in natural image classification, but at the same time ViL beats them in structured tasks like semantic segmentation. This highlights the strengths and weaknesses of the respective methods, which we believe is a valuable insight. In contrast, other vision sequence models (e.g, [2, 3, 4]) simply omit DeiT-II/DeiT-III baseline models, concealing the fact that sequence models are not at the performance of optimized transformers in natural image classification on larger scales yet. We refrain from this practice and instead opt for transparency by comparing even against the most optimized state-of-the-art models instead of "outdated" ones.
>
>
>
> [1] Dosovitsky 2020, "An Image is Worth 16x16 Words: Transformers for Image Recognition at Scale" https://arxiv.org/abs/2010.11929
>
> [2] Zhu 2024, "Vision Mamba: Efficient Visual Representation Learning with Bidirectional State Space Model" https://arxiv.org/abs/2401.09417
>
> [3] Wang 2024, "Mamba-R: Vision Mamba ALSO Needs Registers" https://arxiv.org/abs/2405.14858
>
> [4] Duan 2024, "Vision-RWKV: Efficient and Scalable Visual Perception with RWKV-Like Architectures" https://arxiv.org/abs/2403.02308
>
>
>
> **Scaling to larger model sizes**
>
> Given the scaling curves of the xLSTM in language modeling, we see no reason why ViL would not scale to larger scales. However, as there is currently no efficient hardware implementation of the mLSTM available, training is still quite slow which makes larger scales expensive. Note that also early works of transformers (e.g., DeiT) or mamba (e.g., Vim) did not train extremely large models. As implementations got better and compute got cheaper, it made training of larger models more accessible which made follow-up works also train larger models (e.g., DeiT-III, Mamba-R).
>
>
>
> **FLOPS in recurrent models**
>
> While we do agree that FLOPS are not a perfect measure to compare runtime it is the currently most sensible metric for computational effort due to the (purely technical) limitations of the lack of an optimized hardware implementation of the mLSTM, as discussed throughout the paper. However, we would like to point out that the mLSTM is fully parallelizable and can therefore operate in a fully parallel way and completely omit any recurrence within the model. Given an efficient hardware implementation, FLOPS should therefore be highly correlated with runtime, as also discussed in the general response. Note that we do not use any vanilla LSTM, or sLSTM layers, which would be much slower due to the recurrence relation mentioned by the reviewer.
>
> Notably, ViL is already much faster than Vim, as shown in Appendix A.1, despite Vim using a custom CUDA implementation.
>
>
> **Feature downsampling in ViL**
>
> ViL uses an isotropic architecture, that uses the same internal resolution (16x16 patches correspond to 1 patch token) for all layers, without any downsampling operations in between. We added this information to the caption of Figure 1.

---

> > ### Comment · Reviewer_VTdU · 2024-11-28
> > **Thanks for the response**
> >
> > I appreciate the authors' response to my questions. Some of my concerns are addressed but the problem of scalability remains. LSTM models scaling well in NLP does not necessarily means it also scales well in vision. Some of the responses seem contradictory: the authors claim that ViL is significantly faster than vision Mamba models but cannot provide results for larger size models due to "training is still quite slow". Given Mamba-R can scale to the large size (>300M params), it should not be a big challenge for hardwares.
> >
> > As my concern remains, I keep my original rating.

---

> > > ### Author Response · Authors · 2024-11-28
> > >
> > > Thank you for your response!
> > >
> > > We would like to clarify your point regarding training a ViL-L model.
> > >
> > > Mamba-R does not report any runtimes throughout their paper. Vision-Mamba only reports misleading runtime numbers, i.e., they compare their memory and compute requirements in fp32 with their CUDA kernel against transformers without FlashAttention and in fp32. We understand that this can give the impression that Mamba based backbones are rather efficient, which is not the case. We tried to reproduce the runtime and memory consumption plots of Vim where we got similar results to their reported ones in fp32 with their CUDA kernel. Training in fp32 is extremely expensive and is practically irrelevant as mixed precision training greatly speeds up training for ViTs while preserving performance. When training ViT and Vim in fp16 (the actual setting in which ViTs are trained), ViTs are 8x faster than Vim.
> > >
> > > We benchmark Vim with their original implementation and CUDA kernel and compare it against Vision-LSTM in Appendix A.1. Using the runtime numbers for Vim-L from this benchmark and extrapolating it to a 300 epoch ImageNet-1K training, it would result in 2300 GPU-hours that are required to train a mamba based 300M parameter model. While the current implementation of ViL would only require 1400 GPU-hours, it is still a large amount of compute to train a single model. Additionally, a single training run does not suffice as important hyperparameters (e.g., learning rate) need to be tuned per model size to get good results. For reference, 2300 GPU-hours amount to 12 days of training on a single 8xA100 node.
> > >
> > > Therefore, while the Mamba-R authors had a big enough compute budget to train a 300M parameter model, the amount of GPU-hours required to train such a model are vast. While we understand that this is not obvious from their paper, as they completely omitted discussion on the large compute efforts expended, we would argue that not having the compute budget to train ViL-L in its current implementation should not be seen as a major limitation as getting a sufficiently large compute budget is extremely difficult and mostly outside of our control.
> > >
> > > We believe that it is important to discuss the current limitation of vision sequence models, namely that they are still quite slow compared to ViTs or CNNs. While other works decided to omit discussion on this big weakness, we instead aim to make efficient modeling choices (e.g., alternating block design) and put our approach into perspective by comparing FLOPS against all models, runtimes against other sequence vision models and discussing the fact that transformers are still faster due to optimized hardware implementations. Note that the mLSTM will be faster than transformers once we have an optimized hardware implementation.

---

### Author Response · Authors · 2024-11-18
**General Response**

We thank all reviewers for their valuable feedback and for their constructive comments.

We are glad that the reviewers found our work valuable and highlighted our creative modeling appraoch, clear explanation of the architecture and the extensive experimental evaluation.


**Paper updates**

We integrated various suggestions of the reviewers into the paper:

- Expanded the limitations in Section 5 to integrate related work to justify that the mLSTM will be faster than transformers once an optimized hardware implementation is available.
- Added clarifications whether downsampling layers are used in Figure 1 (reviewer VTdU).
- Restructured the mLSTM background section to put it better into the context of vision sequence models and added additional context for the role of the mLSTM to Figure 1 (reviewer xJrX).



**Lack of optimized hardware implementation**

A shared concern among reviewers is the lack of an optimized hardware implementation to greatly speedup computations, which would enable further hyperparameter optimization, even more extensive experiments, validation of model scaling laws on larger models, training on larger datasets such as ImageNet-21K and fully leveraging the linear complexity by training models for tasks that greatly benefit from higher resolutions. We agree with the reviewers that this is a limiting factor and also discuss this limitation in our paper. However, we want to stress that developing a optimized hardware implementation is not something trivial and requires expert domain knowledge in specialized programming languages as well as a deep understanding of numerical precisions, hardware layouts, parallel processing, compute to memory interactions and many more areas. This is underlined by the fact that optimized hardware implementations are often standalone papers (e.g., [1, 2]) at major conferences where the first optimized hardware implementation of the attention mechanism came out 5 years after the introduction of the transformer. We have been actively working on an efficient hardware implementation, but the workload is far beyond the scope of this work and it would also not change any of the presented results.

Additionally, we already made a significant effort to make our architecture as efficient as possible, using the tools that are currently available to us. Notably, our architecture is already much faster (up to 70\%) than Vision-Mamba (Vim) despite Vim using a custom CUDA kernel, as shown in Appendix A.1. For reference, in language modeling, Mamba is roughly on-par with transformers in terms of speed and 4x faster than than the xLSTM (as mentioned in [3]), again, due to the current lack of efficient hardware implementation of the mLSTM. These considerations further underline the potential of our simple and efficient design for vision applications.

While the Vision-LSTM has still longer runtimes than Vision-Transformers (due to transformers being 7 years old with many optimization iterations), the results of recent linear attention mechanisms show impressive FLOPS utilization (e.g., [4]). As the mLSTM can be parallelized with similar techniques it is only a matter of time that the mLSTM achieves a similar FLOPS utilization, which will make the mLSTM faster than transformers once an efficient hardware implementation is available.


We would argue that our contributions to efficiently adapt the mLSTM to computer vision and show its potential on various tasks are sufficiently important to provide valuable insights to the community and open new avenues for potential applications. We also want to reiterate that an efficient hardware implementation would not change any of the insights of our paper, as it will only make Vision-LSTM faster.


[1] Dao NeurIPS 2022, "FlashAttention: Fast and Memory-Efficient Exact Attention with IO-Awareness" https://arxiv.org/abs/2205.14135

[2] Dao ICLR 2024, "FlashAttention-2: Faster Attention with Better Parallelism and Work Partitioning" https://arxiv.org/abs/2307.08691

[3] Beck NeurIPS 2024, "xLSTM: Extended Long Short-Term Memory" https://arxiv.org/abs/2405.04517

[4] Yang 2023, "Gated Linear Attention Transformers with Hardware-Efficient Training" https://arxiv.org/abs/2312.06635

---

### Meta-Review · Area_Chair_m94m · 2024-12-21

**Metareview:**

The authors propose a novel recurrent architecture for vision tasks, Vision-LSTM, which is an extension of the xLSTM model from language to vision.

Reviewers had some concerns about the novelty of the approach, since it is largely an application of xLSTM from language. Similar to Vision Transformers, the image is tokenised by patchifying it, allowing one to apply the same architecture from language. Results on ImageNet, ADE20k semanatic segmentation and VTAB transfer learning were largely promising, though some reviewers felt that the authors should have done more, and especially larger-scale experiments. This is particularly the case because Vision-LSTM has linear computational and memory complexity with respect to the sequence length unlike transformers, but the authors do not do experiments in scenarios which would make proper use of this.

Although some reviewers complained about the runtime and the lack of optimised implementations for Vision-LSTM, the AC is cognizant of the fact that transformers have been engineered for several years to become as fast as they are now. And so it is not so feasible for the authors to have this straight away.

On the balance, the AC and Senior-AC feel that this paper could have a postive impact on the community, and encourage more work on recurrent architecture for vision. Therefore, the decision is to accept the paper.

**Additional Comments On Reviewer Discussion:**

Please see above. Reviewers had some concerns about the novelty of the approach, since it is largely an application of xLSTM from language. Similar to Vision Transformers, the image is tokenised by patchifying it, allowing one to apply the same architecture from language. Results on ImageNet, ADE20k semanatic segmentation and VTAB transfer learning were largely promising, though some reviewers felt that the authors should have done more, and especially larger-scale experiments. This is particularly the case because Vision-LSTM has linear computational and memory complexity with respect to the sequence length unlike transformers, but the authors do not do experiments in scenarios which would make proper use of this.

Although some reviewers complained about the runtime and the lack of optimised implementations for Vision-LSTM, the AC is cognizant of the fact that transformers have been engineered for several years to become as fast as they are now. And so it is not so feasible for the authors to have this straight away.

---

### Decision · Program_Chairs · 2025-01-22

Accept (Poster)